

**The role of dew as a nighttime reservoir and morning**
**source for atmospheric ammonia**
**Gregory R. Wentworth[1], Jennifer G. Murphy[1], Katherine B. Benedict[2], Evelyn J.**
**Bangs[2], and Jeffrey L. Collett Jr.[2]**
[1]{Department of Chemistry, University of Toronto, 80 St. George Street, M5S 3H6, Toronto,
ON, Canada}
[2]{Department of Atmospheric Science, Colorado State University, 3915 W. Laporte Ave.,
80523, Fort Collins, CO, USA}
Correspondence to: J. G. Murphy (jmurphy@chem.utoronto.ca)
**Abstract**
Several field studies have proposed that the volatilization of $NH_3$ from evaporating dew is
responsible for an early morning pulse of ammonia frequently observed in the atmospheric
boundary layer. Laboratory studies conducted on synthetic dew showed that the fraction of
ammonium ($NH_4^+$) released as gas-phase ammonia ($NH_3$) during evaporation is dependent on
the relative abundances of anions and cations in the dew. Hence, the fraction of $NH_3$ released
during dew evaporation ($Frac(NH_3)$) can be predicted given dew composition and pH. Twelve
separate ambient dew samples were collected at a remote high elevation grassland site in
Colorado from 28 May to 11 August, 2015. Average $[NH_4^+]$ and pH were 26 μM and 5.2,
respectively, and were on the lower end of dew $[NH_4^+]$ and pH observations reported in the
literature. Ambient dew mass (in g $m^{-2}$) was monitored with a dewmeter, which continuously
measured the mass of a tray containing artificial turf representative of the grass canopy to track
the accumulation and evaporation of dew. Simultaneous measurements of ambient $NH_3$
indicated that a morning increase in $NH_3$ was coincident in time with dew evaporation, and that
either a plateau or decrease in $NH_3$ occurred once the dew had completely evaporated. This
morning increase in $NH_3$ was never observed on mornings without surface wetness (neither
dew nor rain, representing one-quarter of mornings during the study period). Dew composition
was used to determine an average $Frac(NH_3)$ of 0.94, suggesting that nearly all $NH_4^+$ is released



back to the boundary layer as $NH_3$ during evaporation at this site. An average $NH_3$ emission of
6.2 ng m$^{-2}$ s$^{-1}$ during dew evaporation was calculated using total dew volume ($V_{dew}$) and
evaporation time ($t_{evap}$), and represents a significant morning flux in a non-fertilized grassland.
Assuming a boundary layer height of 150 m, the average mole ratio of $NH_4^+$ in dew to $NH_3$ in
the boundary layer at sunrise is roughly $1.6 \pm 0.7$. Furthermore, the observed loss of $NH_3$ during
nights with dew is approximately equal to the observed amount of $NH_4^+$ sequestered in dew at
the onset of evaporation. Hence, there is strong evidence that dew is both a significant night-
time reservoir and strong morning source of $NH_3$. The possibility of rain evaporation as a source
of $NH_3$, as well as dew evaporation influencing species of similar water solubility (acetic acid,
formic acid, and HONO) is also discussed. If release of $NH_3$ from dew and rain evaporation is
pervasive in many environments, then estimates of $NH_3$ dry deposition and $NH_x$ ($\equiv NH_3 + NH_4^+$)
wet deposition may be overestimated by models that assume that all $NH_x$ deposited in rain and
dew remains at the surface.

## 14   1   Introduction

Ammonia ($NH_3$) is the most prevalent alkaline gas in the atmosphere and has important
implications for both climate and air quality (Seinfeld and Pandis, 2006). For instance, $NH_3$
partitions to acidic fine particulate matter ($PM_{2.5}$, aerosol with diameter < 2.5 μm) to form
particulate-phase ammonium ($NH_4^+$), which alters aerosol properties such as hygroscopicity
(Petters and Kreidenweis, 2007) and scattering efficiency (Martin et al., 2004). High
atmospheric loadings of $PM_{2.5}$ can lead to adverse health effects (Pope et al., 2002) as well as
reduced visibility. $NH_3$ is primarily emitted to the atmosphere through agricultural activities
(e.g. fertilization, animal husbandry) in addition to natural sources (e.g. soil, vegetation, oceans,
volcanoes, wildfires) and other anthropogenic sources (vehicles and industry) (Reis et al.,
2009). Deposition of atmospheric $NH_x$ ($\equiv NH_3 + NH_4^+$) can cause eutrophication and soil
acidification in sensitive ecosystems (Krupa, 2003). Hence, there is great interest in being able
to accurately model sources, sinks and reservoirs of $NH_x$.
A common feature in the diurnal cycle of atmospheric $NH_3$ mixing ratios is a morning increase
or "spike" that typically occurs around 7:00-10:00. Frequently observed in many environments,
current hypotheses to explain the morning $NH_3$ increase include dew evaporation (Gong et al.,
2011; Wentworth et al., 2014; Wichink Kruit et al., 2007), plant and/or soil emissions (Bash et
al., 2010; Ellis et al., 2011), mixing down of $NH_3$-rich air during the break-up of the nocturnal
boundary layer (Walker et al., 2006) and automobile emissions during morning rush hour (Gong



et al., 2011; Löflund et al., 2002; Nowak et al., 2006; Whitehead et al., 2007). Several field
studies have indicated that $NH_3$ desorption from microscopic water films on leaf surfaces can
yield significant fluxes (Burkhardt et al., 2009; Flechard et al., 1999; Neirynck and Ceulemans,
2008; Sutton et al., 1998); therefore, it is reasonable to suggest that macroscopic dew droplets
have the same effect. Wentworth et al. (2014) observed a larger morning increase following
nights with high relative humidity (RH, a surrogate for dew) and Wichink Kruit et al. (2007)
found increasing upward fluxes as soon as the canopy began to dry as measured by a leaf
wetness sensor.
Dew generally forms during calm, clear nights when radiative cooling of the surface favours
the condensation of water (Richards, 2004). Formation typically starts shortly after sunset and
lasts until sunrise when surface heating and a drop in RH initiate evaporation. Over the last five
decades, several dozen studies have characterized dew composition and have found that $NH_4^+$
is a ubiquitous constituent of dew and, in some environments, can be the most abundant cation
(e.g. Polkowska et al., 2008; Wagner et al., 1992; Yaalon and Ganor, 1968; Yadav and Kumar,
2014). Average $[NH_4^+]$ reported in dew ranges from 25 μM (Lekouch et al., 2010) to 1600 μM
(Yadav and Kumar, 2014). The composition of dew is primarily controlled by dissolution of
water soluble gases (e.g. $NH_3$, $HNO_3$, $CO_2$, $SO_2$) and deposition of coarse mode particles (larger
than $PM_{2.5}$ but smaller than 10 μm in diameter) (Takeuchi, 2003).
Field-scale models typically allow $NH_3$ to only deposit on leaf cuticles (i.e. it cannot desorb)
and use an empirically-derived function of RH and cuticle acidity to calculate a cuticle
deposition velocity. This parameterization accounts for enhanced deposition to acidic water
films on leaf surfaces. There are only a handful of field-scale $NH_3$ models that allow for
desorption of $NH_3$ from drying water films on leaf cuticles (Burkhardt et al., 2009; Flechard et
al., 1999; Neirynck and Ceulemans, 2008; Sutton et al., 1998). Three of these studies (Flechard
et al., 1999; Neirynck and Ceulemans, 2008; Sutton et al., 1998) compared models with and
without cuticle desorption and found that allowing for $NH_3$ emission from water films on
cuticles agrees better with observed fluxes during the morning. Other field-scale measurements
attribute discrepancies between measured and modelled morning fluxes to $NH_3$ emission during
the drying of canopies (e.g. Bash et al., 2010; Walker et al., 2013; Wyers and Erisman, 1998).
Most larger scale (regional or global) chemical transport models (CTMs) still employ highly
simplified deposition schemes for $NH_3$ using look-up tables for deposition velocity and canopy





resistance terms (Wesely, 1989). In other words, they treat deposition and emission of $NH_3$
independently despite abundant field evidence that these processes are often highly coupled.
Although most $NH_3$ surface-air exchange studies account for enhanced deposition to
microscopic water films and several even model $NH_3$ desorption, far fewer have attempted to
assess the role that macroscopic dew has on influencing $NH_3$ fluxes. This partly stems from the
inherent difficulty in measuring dew amount, composition, and pH. Only a few $NH_3$ surface-
air exchange studies have attempted to measure dew composition and pH, and did so by
manually collecting enough individual droplets in pipettes to perform bulk analyses (Bash et
al., 2010; Burkhardt et al., 2009; Walker et al., 2013). To constrain dew amount, Burkhardt et
al. (2009) used an empirically-derived relationship to approximate water film thickness from a
leaf wetness sensor. Walker et al. (2013) estimated dew amount by difference in water mass
between wet and dried leaves. Both studies acknowledge the large uncertainties associated with
these methods for estimating total dew amount in the canopy. Regardless, Walker et al. (2013)
estimated a maximum flux of 17.6 ng $NH_3$ $m^{-2}$ $s^{-1}$ from dew evaporation in a fertilized corn
canopy. One key assumption in this calculation is that all of the $NH_4^+$ present in dew is released
as $NH_3$ during evaporation.
Few studies have examined the fate of semi-volatile solutes (e.g. $NH_4^+/NH_3$, $NO_2^-/HONO$,
acetate/acetic acid etc…) in rain, cloud, fog or dew during droplet evaporation. Takenaka et al.
(2009) studied the chemistry of drying aqueous salts in a series of lab experiments and found
that the fraction of "volatile" anions (which they operationally defined as $NO_2^-$, acetate, and
formate) remaining on the surface as a salt upon evaporation depends on the relative equivalents
of "non-volatile" cations ($Na^+$, $K^+$, $Mg^{2+}$ and $Ca^{2+}$) and "non-volatile" anions ($Cl^-$, $NO_3^-$, and
$SO_4^{2-}$). The fraction of volatile anion ($X^-$) that is released during evaporation (as $HX_{(g)}$) can then
be predicted using the following equation (Takenaka et al., 2009):
$$Frac(X) = \frac{[X]_i - (\Sigma cations - \Sigma anions)}{[X]_i} \qquad (1)$$
where Frac(X) is the fraction of the initial anion released to the atmosphere during evaporation,
$[X]_i$ is the initial equivalents of "volatile" anion $X$, and $\Sigma$cations and $\Sigma$anions are the sums of
"non-volatile" cations and anions, respectively. The authors performed numerous experiments
for $NO_2^-$, acetate, and formate under a wide range of solute concentrations and pH values and
found that Eq. (1) was consistently able to predict the fraction of each constituent liberated
during evaporation of aqueous salt solutions. Although not the focus of the work, Takenaka et





al. (2009) also performed some evaporation experiments on solutions containing $NH_4^+$ and were
able to predict Frac($NH_3$) with an analogous equation:
$$Frac(NH_3) = \frac{[NH_4^+]_i - (\Sigma anions - \Sigma cations)}{[NH_4^+]_i}$$   (2)
where $[NH_4^+]_i$ is the initial ammonium concentration in the solution.
Few field studies have simultaneously quantified both dew and atmospheric composition. He
et al. (2006) observed HONO emission from a drying forest canopy and performed lab studies
to show that, on average, ~90% of $NO_2^-$ was released as HONO during droplet evaporation.
Rubio et al. (2009, 2012) found positive correlations between formaldehyde, phenols and
HONO in dew and the atmosphere. However, none of these studies, or those mentioned earlier
for $NH_3$, accurately measured dew amount (in g m$^{-2}$) on the surface, so the relative abundances
of the analyte in the dew and gas-phase could not be reliably calculated.
Therefore, great uncertainty exists regarding the influence of dew on boundary layer
composition, particularly with respect to $NH_3$ mixing ratios. Motivated by the paucity of data
on dew-atmosphere $NH_3$ fluxes, as well as uncertainties about the origin(s) of the frequently
observed yet currently unexplained morning $NH_3$ spike, the specific goals of this study are to:
1) Determine the fate of $NH_4^+$ during dew evaporation (Section 3.1). What is the ratio of
$NH_x$ released as $NH_3$ versus $NH_4^+$ remaining on the surface as a non-volatile salt? What
factor(s) govern this ratio?
2) Simultaneously quantify dew amount, $NH_3$ mixing ratio, and dew composition at the
onset of evaporation at a field site (Section 3.2).
3) Use measurements from 1) and 2) to calculate the relative abundance of $NH_4^+$ in dew
and $NH_3$ in the boundary layer, as well as $NH_3$ fluxes from dew evaporation (Section
3.3). Is dew a significant night-time sink or reservoir for $NH_3$? Is $NH_3$ release from dew
an important morning source?
4) Evaluate if $NH_3$ is also released during rain evaporation (Section 3.4).
5) Assess the impact of dew evaporation for other water-soluble gases (Section 3.5).



**2    Materials and Methods**
**2.1    Drying Chamber**
A drying chamber was used to determine the fraction of $NH_4^+$ lost as $NH_3$ during droplet
evaporation and was based on the set-up used by Takenaka et al. (2009). The set-up consists of
a zero air cylinder (AI Z300, Airgas) and mass flow controller which deliver zero air at a
controlled flow rate into a drying chamber (URG-2000-30H, URG Corp.) containing droplets
of synthetic dew. Downstream of the drying chamber is an annular denuder (URG-2000-30,
URG Corp.) coated with a phosphorous acid solution (10 g $H_3PO_3$ in 100 mL deionized water
and 900 mL HPLC grade methanol) to capture any $NH_3$ emitted during dew drying.
At the beginning of each experiment, 26 droplets (20 μL each) of synthetic dew were deposited
in the drying chamber and dried over the course of several hours by exposure to a flow of 2 L
$min^{-1}$ of zero air. Immediately after the last droplet had dried, the residue remaining in the
chamber was extracted twice using two separate 10 mL aliquots of deionized water (18.2 MΩ
$cm^{-1}$) and vigorous washing. The second aliquot always contained <10% of each analyte
relative to the first aliquot. The annular denuder was extracted by adding 10 mL of deionized
water and rotating for 10 minutes. Concentrations of ions in all three extracts were quantified
using ion chromatography (IC) systems (DX-500, Dionex Inc.) and an isocratic elution scheme
(1.8/1.7 mM $Na_2CO_3/NaHCO_3$ solution for anions and 0.020 mM methanesulphonic acid
solution for cations). The pH of the dew was determined using a commercial pH meter (Orion
Model 250A, Thermo Scientific). The fraction of each analyte remaining in the salt residue was
then calculated, as well as the fraction of ammonium lost as $NH_3$ based on the total $NH_x$ amount
measured in the three aliquots.
Experimental parameters (composition, pH and drying time) were varied to determine the
factor(s) responsible for the fraction of $NH_3$ that is released from dew as it dries. Synthetic dew
was prepared by dissolving salts in deionized water to the desired concentration. All salts were
reagent grade, obtained from Sigma Aldrich and used without further purification. The pH was
then adjusted with either concentrated acid (HCl) or base (NaOH). A total of nine different
synthetic dews were prepared to mimic ambient dew composition reported from previous
studies (e.g. Lekouch et al., 2010; Takenaka et al., 2003; Yadav and Kumar, 2014). The pH and
concentrations of the nine synthetic dews are listed in supplemental Table S1.





Synthetic dew was deposited as 20 µL droplets, which corresponds to a hemi-spherical diameter
of ~4.25 mm. Takeuchi et al. (2002) found that the diameter of most dew droplets range from
0.8 to 1.0 mm in diameter; however, applying such small droplets would bring the concentration
of the extracts below detection limit. In order to maintain solute concentrations relevant to
ambient dew, but generate sufficient signal for analysis, it was necessary to use 20 µL droplets.
The impact of larger droplet size on $NH_3$ liberation was tested by performing several drying
experiments on four 140 µL drops (~8.1 mm in diameter). These larger droplets had no effect
on the fraction of $NH_3$ emitted relative to the 20 µL droplets.

## 2.2 Field Site

Ambient measurements of dew composition, dew volume and gas-phase $NH_3$ were obtained at
a field site situated on the eastern edge of Rocky Mountain National Park (RMNP) in Northern
Colorado (40.2783° N, 105.5457° W; 2784 m a.s.l.) from 28 May to 31 August, 2015. The field
site is remote with the nearest town (Estes Park, CO, population ~6,000) located approximately
14 km north. This site is also used by the Interagency Monitoring of Protected Visual
Environments (IMPROVE) and EPA Clean Air Status and Trends Network (CASTNet)
programs for air quality monitoring and has been the location of extensive studies on nitrogen
deposition (Beem et al., 2010; Benedict et al., 2013a) and atmospheric reactive nitrogen
(Benedict et al., 2013b). The field site is a grassland clearing approximately 150 m in diameter
surrounded by a mixed aspen and pine forest (average summertime maximum leaf area index
of 1.5). In addition, excessive nitrogen deposition at RMNP has been linked to ecological
impacts including changes in diatom assemblages (Baron, 2006; Wolfe et al., 2003) and shifts
in a dry alpine meadow community (Bowman et al., 2012). Recently, Nanus et al. (2012)
suggested that the critical load for nitrogen deposition (a value beyond which negative
ecological impacts are observed) has been exceeded in ~21% of the Rocky Mountains. The
existing body of knowledge regarding reactive nitrogen at RMNP makes this site ideal to
examine how dew-atmosphere interactions affect $NH_3$ in the boundary layer as well as its
deposition.

## 2.3 Atmospheric Measurements

$NH_3$ was measured using a Picarro G1103 Analzyer, a cavity ringdown spectroscopy
instrument. The inlet line was 3.56 cm diameter Teflon tubing located approximately 2.5 m
above ground level. The entire length of the 0.61 m inlet line was insulated and heated to 40°C





to minimize wall losses. A filter (Picarro P/N S1021) was placed on the end of the inlet to
prevent particles from entering the instrument. The filter was also heated which may have
caused $NH_4NO_3$ to volatilize from the filter or air stream resulting in an overestimation of the
ammonia concentration. However, a previous study at the site found that, on average, only a
small fraction of particulate $NH_4^+$ exists as $NH_4NO_3$ during the summer (Benedict et al., 2013b).
Furthermore, the same study found that $NH_3$ was the majority of the $NH_x$ ($\equiv NH_3 + NH_4^+$)
loading. Hence, it is unlikely that there is a large interference from $NH_4NO_3$ volatilization.
Calibrations were performed twice during the field deployment using MKS mass flow
controllers, a certified 2 ppm $NH_3$ cylinder (AirGas), and a zero air source (Teledyne Zero Air
Generator Model 701). The calibration gas was split between the Picarro and a phosphorus acid
(10% w/v) coated denuder to act as a check of the concentration. The denuder was sampled at
2 L min$^{-1}$ and the total volume was recorded using a dry gas meter. The concentration
determined by the denuder was used as the "true" concentration in the calibration curve.
Meteorological measurements were made at the site by a 10 m tower operated by the National
Park Service. Measurements are reported at 1 hour intervals for solar radiation, temperature,
wind speed, wind direction, standard deviation of the wind direction over the period, relative
humidity, and rainfall.

## 2.4 Dew Measurements

Ambient dew samples at RMNP were gathered using a dew collector with a design similar to
Guan et al. (2014). The collector was built in-house and consists of a wooden base that supports
a 7 cm-thick polystyrene foam block with an area of 48 x 60 cm. The top surface of the
polystyrene block is covered by a 0.2 mm-thick polytetrafluoroethylene (Teflon®) sheet. The
Teflon® sheet is parallel to the ground at a height of 30 cm. During the night the Teflon® sheet
undergoes radiative cooling while the polystyrene insulates the sheet from below. This results
in dew formation on the Teflon® surface which can be manually collected into clean sample
bottles the following morning using a pre-cleaned scraper and funnel. The emissivity of Teflon®
is 0.94 (Baldridge et al., 2009) and is very similar to that of vegetation (0.95) (Guan et al.,

28 2014).

The dew collector was deployed before dusk on nights that had a forecast favourable for dew
formation (high relative humidity, light winds, and clear skies). The Teflon® surface was
cleaned immediately before deployment in a two-step process: 1) splashing ~1 L of deionized



water across the surface, followed by 2) squirting ~30 mL of deionized water on the surface
and scraping it off using a plastic scraper. The latter step was repeated 10 times, and the 10th
rinse was collected and used as a field blank for dew collected the following morning. Prior to
dew collection, the funnel and scraper were rinsed 10 times with deionized water. This cleaning
procedure proved sufficient and is similar to prior studies using a similar collector (e.g. Okochi
et al., 2008; Wagner et al., 1992). Dew samples were collected into 15 mL polypropylene
sample bottles in order to minimize headspace during transport and storage.
Chemical analyses of all dew samples were performed within 6 hours of collection, with the
exception of one sample which was stored at 4 °C and analysed 48 hours later. The
concentration of ions ($Na^+$, $NH_4^+$, $K^+$, $Mg^{2+}$, $Ca^{2+}$, $Cl^-$, $NO_2^-$, $NO_3^-$, $SO_4^{2-}$, $PO_4^{3-}$, acetate, formate
and oxalate) in dew samples was determined through ion chromatography and pH was
measured with a pH meter, as outlined in section 2.1. The total organic carbon (TOC) and
inorganic carbon (IC) were quantified with a commercial TOC analyser (TOC-$V_{CSH}$, Shimadzu
Corp.) equipped with a total nitrogen (TN) analyser (TNM-1, Shimadzu Corp.) for
quantification of TN. Concentrations of analytes in ambient dew samples were background
corrected by subtracting the volume-weighted concentration in the tenth rinse collected the prior
evening which is likely an upper bound for the background signal given that some volatile
solutes will be scavenged from the air during application and collection of the rinse.
It was also necessary to quantify the volume of dew ($V_{dew}$) that formed each night. The dew
collector is not suitable since $V_{dew}$ obtained from the collector is not necessarily representative
of $V_{dew}$ that forms naturally on the grassland canopy at RMNP. Numerous methods and
instruments exist to measure $V_{dew}$; for instance, the cloth-plate method (Ye et al., 2007),
lysimeter-related instruments (Grimmond et al., 1992; Price and Clark, 2014), and eddy-
correlation techniques (Moro et al., 2007). Although there is no standard method to measure
$V_{dew}$, Richards (2004) provides a detailed overview of various techniques that have been used
to collect and quantify dew.
For this study, we constructed a dewmeter similar to that of Price and Clark (2014). The design
consists of a circular collection tray (diameter of 35 cm) that is attached to the top of an
analytical balance (HRB 3002, LWC Measurements). The balance has a resolution of 0.01 g
and a maximum load of 3000 g. The tray contains artificial turf that is intended to be
representative of the grass at the RMNP field site during the early part of the growing season.
The balance was contained in a weatherproof box with a hole cut in the lid to accommodate the





tray/turf. The mass on top of the balance was recorded to a laptop at a rate of 5 Hz so that the
mass of dew was continuously monitored as it formed and evaporated. The data were averaged
to 10 min to achieve better signal-to-noise.
Price and Clark (2014) performed an extensive characterization of the dewmeter and compared
dew formation/evaporation on co-located dewmeters containing real turf and artificial turf. The
authors found that $V_{dew}$ and the dew deposition rate were identical between the two turfs. In
other words, the radiative properties and surface area of artificial turf sufficiently mimic real
turf such that artificial turf can be used as a surrogate to quantify $V_{dew}$ and its temporal
evolution. The advantage of using artificial turf is that there are no changes in mass due to
evapotranspiration during the daytime. The dewmeter is also capable of quantifying rainfall and
its evaporation. However, if the rainfall is too intense (≥2 mm) then the tray becomes flooded
and must be replaced with a dry tray/turf.
**2.5  Flux Calculation**
$NH_3$ fluxes from dew evaporation were calculated using the following equation:

$$F_{NH3} = \frac{[NH_4^+] \cdot V_{dew}}{t_{evap}} \cdot Frac(NH_3) \cdot 17{,}031 \qquad (3)$$

where $F_{NH3}$ is the average emission flux (in ng m$^{-2}$ s$^{-1}$) during dew drying, $[NH_4^+]$ is the
concentration of ammonium in dew (in μM), $V_{dew}$ is the volume of dew in the canopy (in L m$^{-2}$
), $t_{evap}$ is the time it takes for dew to evaporate (in s), $Frac(NH_3)$ is the fraction of $NH_4^+$ in the
dew that is released as $NH_3$, and 17,031 is to convert μmol to ng. It is important to note that Eq.
(3) yields the average $F_{NH3}$ during evaporation and cannot account for any variations in $F_{NH3}$
over the evaporation period. The dewmeter was used to record $V_{dew}$ and $t_{evap}$, whereas sample
from the dew collector was used to quantify $[NH_4^+]$ and calculate $Frac(NH_3)$. The dewmeter is
automated and was deployed continuously from 22 June until 31 August (and intermittently
between 27 May and 21 June), whereas the dew collector requires manual cleaning and
collection so was only deployed when forecasts were favourable for dew formation.



**3    Results and Discussion**
**3.1    Fraction of NH$_3$ that evaporates from drying dew**
We tested the validity of Eq. (2) by performing a series of drying experiments similar to
Takenaka et al. (2009) but specifically targeting conditions relevant for dew (i.e. composition
and drying time). Takenaka et al. (2009) used solutions in the mM range with drying times of
~9 h, whereas natural dew is typically less concentrated (μM range) and usually dries within a
few hours. The composition of synthetic dew (Table S1) and drying time (~2.5 h) in this work
are a better representation of natural dew.
Figure 1 shows the measured Frac(NH$_3$) versus predicted Frac(NH$_3$) from an updated form of
Eq. (2) (see below for details) for the nine synthetic dews. Drying experiments were performed
three times per dew composition, and error bars in Fig. 1 denote the standard deviation between
experiments. The amount of NH$_x$ ($\equiv$NH$_4^+$ + NH$_3$) recovered was always within 20% of the
amount of NH$_4^+$ added at the beginning of the experiment. There is good agreement between
the measured and predicted Frac(NH$_3$) which is mostly consistent with the findings of Takenaka
et al. (2009) with a few key differences: 1) the majority of acetate and formate remained as a
salt after evaporation, 2) HCO$_3^-$ was an important constituent in the anion balance, and 3) the
pKa of each substance must be considered. Although acetic acid, formic acid, and carbonic acid
are relatively volatile, the conjugate bases can (and do) form non-volatile salts upon evaporation
if there is an excess of cations. Furthermore, if the pH is near or less than the pKa of the acids
then a significant fraction will be neutral (protonated) and unable to form a salt. Hence, we
update the definition of Σanions in Eq. 2 to include acetate, formate, and bicarbonate (also
reflected in Fig. 1) which yields much better agreement in predicted versus measured
Frac(NH$_3$).
Since ion chromatography quantifies the total amount of each species (i.e. both charged and
neutral forms) it is necessary to use pH and the acid dissociation constant (K$_a$) for each species
to calculate the ionic fraction of each. Furthermore, Takenaka et al. (2009) recommend
including carbonate/bicarbonate in the ion balance for field samples. The authors did not
account for CO$_2$-equilibria since their lab experiments were performed under strict CO$_2$-free
conditions, whereas our synthetic dew samples had sufficient exposure to lab air to equilibrate
with atmospheric CO$_2$ (~500 ppm in the lab) as verified by subsequent inorganic carbon
measurements (section 2.4). Hence, we calculated the amount of HCO$_3^-$ and CO$_3^{2-}$ in synthetic





dew using pH and carbonate equilibria assuming $P_{CO2}$ = 500 ppm. Charge imbalance calculated
in Eq. (2) is a result of $CO_2$ dissolving (or outgassing if a large quantity of bicarbonate/carbonate
salt was added) as well as the addition of HCl or NaOH.

## 3.2   Dew Parameters

A total of 12 dew samples for chemical analysis were collected at RMNP over the study period.
The equivalent concentrations of ions are given in Fig. 2 and TOC, IC, TN, pH and Frac($NH_3$)
in Table 1. Average values of $[NH_4^+]$ in dew found in the literature span several orders of
magnitude ranging from 25 µM in coastal Croatia (Lekouch et al., 2010) to 1600 µM in urban
India (Yadav and Kumar, 2014). Dew at RMNP is at the lower end of this range with median
$[NH_4^+]$ = 28 µM. In general, the concentrations of all species in RMNP dew are lower than most
previous studies (e.g. Singh et al., 2006; Takenaka et al., 2003; Wagner et al., 1992). This is
due to the remoteness of RMNP resulting in low levels of coarse mode aerosol and water-
soluble gases which tend to control the composition of dew via deposition and dissolution
(Takeuchi, 2003; Wagner et al., 1992). The dominant cations in dew at RMNP are $Ca^{2+}$ and
$NH_4^+$. The former is likely from the deposition of coarse mode soil and/or dust particles and the
latter from gas-phase dissolution of $NH_3$. Acetate and formate are the major anions and may be
the result of dissolution of acetic and formic acid (Wagner et al., 1992) and/or the products of
aqueous-phase oxidation of semi-volatile organics (SVOCs, e.g. aldehydes) which has been
observed in cloud and fog water (Herckes et al., 2007, 2013; Munger et al., 1989). The area
surrounding the field site is heavily forested and the boundary layer is likely rich in biogenic
SVOCs which could explain the high TOC content in the dew (average = 6.23 mg C $L^{-1}$). The
ability for dew to act as a medium for aqueous-phase oxidation of SVOCs is outside the scope
of this paper but warrants further investigation.
The average pH of dew at RMNP was 5.19 (median = 5.34) which is on the lower range of what
has been reported for dew. For instance, Yaalon and Ganor (1968) and Xu et al. (2015) found
median dew pH of 7.7 and 6.72 in Jerusalem and Changchun, China, respectively. Whereas
Pierson et al. (1986) reported an average dew pH of 4.0 at a rural site in Pennsylvania in a
region containing several coal-fired power plants. Given the remoteness of RMNP and low
ionic concentrations, $CO_2$ dissolution plays an important role in governing dew pH. Acidic
dews are considered to enhance deposition of $NH_3$ and hinder that of certain weakly acidic
gases (e.g. $SO_2$, organic acids) (Chameides, 1987; Okochi et al., 1996). In addition, the average
summertime $NH_3$ mixing ratio at RMNP is about a factor of 3 higher than that of $HNO_3$





(Benedict et al., 2013b) which is roughly the same ratio as $NH_4^+:NO_3^-$ in dew measured in this
study.
Equation (2) was used to calculate Frac($NH_3$) for ambient dew samples (average = 0.94). Only
three of the twelve samples had a Frac($NH_3$) less than 1 meaning that, in most cases, all of the
$NH_4^+$ present is predicted to volatilize as $NH_3$ during dew evaporation. It is important to note
that acetate, formate, and $HCO_3^-$ were included in the $\sum$anion budget in contrast to Takenaka et
al. (2009). If the aforementioned anions were not included in the Frac($NH_3$) calculation then all
dew samples would have Frac($NH_3$) = 1.
The high Frac($NH_3$) has an important implication for N-deposition: $NH_3$ that is dry deposited
onto a surface wetted with dew does not necessarily contribute to N-deposition. In other words,
$NH_3$ deposited into dew overnight should not necessarily be counted towards the total N-
deposition budget for a given ecosystem. The consequence of this implication likely extends
beyond RMNP and merits additional field measurements of dew to calculate Frac($NH_3$) in other
environments (e.g. agricultural, urban, and rural). To our knowledge, this is the first field study
to quantify the extent to which $NH_4^+$ is released as $NH_3$ during dew evaporation.
## 3.3   Dew-Atmosphere $NH_3$ Fluxes
In this section we examine how the formation and evaporation of dew impacts $NH_3$ in the
boundary layer. Figure 3 shows time series (from 19:00 to 11:00 the following day) of dew
mass (g m$^{-2}$), air temperature (°C) and $NH_3$ mixing ratio (ppbv) on four separate nights with
dew. One feature common to all four panels is the increase of $NH_3$ at the onset of dew
evaporation followed by a plateau or decrease of $NH_3$ once the surface had dried completely.
The features in Fig. 3 are representative of the other 29 nights on which dew formed during the
study period (27 May to 31 August). It should be noted that in Fig. 3c and 3d, the start of the
morning $NH_3$ increase is slightly delayed from the onset of dew evaporation. This may be
attributed to canopy growth over the course of the campaign – during May and June (Figs. 3a
and 3b) the grassland canopy was relatively short (~5 cm) and roughly the same height as the
artificial turf on the dewmeter. However, during July (Fig. 3c) and August (Fig. 3d) the canopy
had grown significantly (up to ~30 cm) providing significant shade to lower parts of the grass
such that dew finished evaporating off the dewmeter prior to complete drying of the canopy.
This would also cause an underestimation of dew amount by the dewmeter towards the end of
the measurement period.





The consistent timing between dew evaporation and the increase in $NH_3$ mixing ratio is strong
evidence that dew evaporation and the early morning $NH_3$ increases are linked, but other
phenomena must be considered. For instance, it is well known that $NH_3$ emissions from plant
stomata and soil are heavily temperature dependent and increase at higher temperatures
(Massad et al., 2010; Sutton et al., 2013; Zhang et al., 2010). However, $NH_3$ decreases after
dew evaporation ceases, despite a continued increase in temperature, suggesting that this
morning increase is not from stomata or soil emissions. Another possible explanation is reduced
deposition after dew evaporation since wet canopies provide a lower resistance to deposition
for water-soluble gases (e.g. $NH_3$) relative to dry canopies (Fowler et al., 2009; Neirynck and
Ceulemans, 2008); however, this scenario requires other continuous source(s) of $NH_3$. If this
were the mechanism responsible for morning $NH_3$ increases then one would expect a plateau
in $NH_3$ after canopy drying. However, Figs. 3a, 3b, and 3d all show $NH_3$ decreases after dew
evaporation. In addition, RMNP is sufficiently remote that morning $NH_3$ increases cannot be
from rush-hour traffic or industrial sources.
It is also useful to consider the behaviour of $NH_3$ on mornings without dew. Of the 72 nights
on which the dewmeter was deployed and functioning, there was night-time rain on 23 of the
nights, and no surface wetness (neither rain nor dew) at sunrise on 16 nights. Typically, dew
formation began around 20:30 and it had completely evaporated by 9:00 the following morning.
Figure 4 compares $NH_3$ mixing ratios from 4:00 to 11:00 on mornings with dew (Fig. 4a) and
without dew or rain (Fig 4b). The clear morning $NH_3$ increase only happens on mornings with
dew, further supporting the hypothesis that dew evaporation has a significant influence on near-
surface $NH_3$ mixing ratios. The traces in Fig. 4 are coloured according to the average $NH_3$
mixing ratio the previous night (from 19:00 to 21:00). The magnitude of the morning increase
is related to the amount of $NH_3$ present the previous night suggesting that most of the $NH_4^+$ in
dew is a result of $NH_3$ dissolution. This is additional evidence that $NH_3$ deposited in dew
overnight at RMNP is recycled back to the atmosphere the following morning upon
evaporation, and should not be counted towards total N-deposition. In other words, the dew
acts as a temporary reservoir for atmospheric ammonia and the cycle of dew formation and
evaporation has a strong influence on boundary layer $NH_3$ concentrations.
Table 1 shows the calculated $NH_3$ fluxes from dew during evaporation (average = 6.2 ng m$^{-2}$ s$^{-}$
$^1$) as well as the relevant parameters required for flux calculations ($t_{evap}$, Frac($NH_3$), and $V_{dew}$).
To our knowledge, only two studies to date have reported $NH_3$ fluxes in a non-fertilized





grassland. Wichink Kruit et al. (2007) used the aerodynamic gradient method to measure a daily
average summertime $NH_3$ flux of 4 ng m$^{-2}$ s$^{-1}$ in a field in the Netherlands, whereas Wentworth
et al. (2014) inferred a daily average soil emission flux of 2.6 ng m$^2$ s$^{-1}$ during August in a rural
field near Toronto, Canada using simultaneous soil and atmospheric measurements and a simple
resistance model. In the context of these previous studies over the same land type, the dew-
related $NH_3$ fluxes at RMNP are significant. Furthermore, it is likely that dew-related $NH_3$
fluxes would be substantially larger at the other field sites given that $NH_3$ mixing ratios were a
factor of 3-10 higher which would result in higher dew $[NH_4^+]$.
For the 12 dew samples listed in Table 1, a simple calculation was performed to estimate the
moles of $NH_4^+$ contained in dew relative to the moles of $NH_3$ in the boundary layer. Particulate
$NH_4^+$ is not considered due to its low mass loadings at RMNP (Benedict et al., 2013b). The
$\mu$mol m$^{-2}$ of $NH_4^+$ in dew at the onset of evaporation was calculated by multiplying $V_{dew}$ by
dew $[NH_4^+]$. One inherent assumption is that $[NH_4^+]_{dew}$ on the collector is representative of the
dew on the dewmeter. An equivalent mole loading (also in $\mu$mol m$^{-2}$) of $NH_3$ in the boundary
layer was calculated by first converting the measured mixing ratio from ppbv to $\mu$mol m$^{-3}$, and
then multiplying by an assumed boundary layer depth of 150 m. The average ratio of
$NH_4^+{}_{,dew}$:$NH_{3,BL}$ is $1.6 \pm 0.7$ for the 12 dew samples collected. In other words, on a per mole
basis there is nearly double the $NH_4^+$ in dew than there is $NH_3$ in a 150 m deep boundary layer.
Unfortunately, there are no measurements at RMNP that allow a better constraint of the
boundary layer height. Assuming a smaller (larger) boundary layer height would increase
(decrease) the $NH_4^+{}_{,dew}$:$NH_{3,BL}$ ratio.
The measured loss of $NH_3$ (in ppbv) during dew nights was used to estimate the sink of $NH_3$
(in $\mu$mol m$^{-2}$) between the onset of dew formation and evaporation. This loss was estimated in
a similar fashion as above assuming: 1) 150 m nocturnal boundary layer, 2) no reactive sinks
(e.g. $NH_4NO_3$ formation), and 3) no exchange with the free troposphere. Figure 5 shows a
correlation plot of estimated $NH_3$ lost on dew nights versus the observed $NH_4^+$ accumulated in
dew. The good correlation and near-unity slope (0.71) show that there is approximate mass
closure between $NH_3$ lost overnight and $NH_3$ sequestered by dew. Although these calculations
are simplistic it is evident that, on average, dew sequesters a significant portion (estimated at
nearly two-thirds) of $NH_3$ over the course of the night. Subsequent studies on dew-atmosphere
interactions should include measurements of boundary layer height so a more thorough mass
balance calculation can be performed.



The loss rate of $NH_3$ on dew nights versus dry nights was examined by fitting the $NH_3$ mixing
ratio to an exponential decay function between 20:00 and 9:00 (or dew evaporation) on the 46
nights in Fig. 4. The fit function used was:
$$[NH_3]_t = [NH_3]_{sunset} e^{-kt} + [NH_3]_{overnight}$$  Eq. (4)
where $[NH_3]_t$ is the mixing ratio of $NH_3$ at time $t$, $[NH_3]_{sunset}$ is the mixing ratio at 20:00,
$[NH_3]_{overnight}$ is the plateau in nocturnal $NH_3$ mixing ratio, and $k$ is an empirical fit parameter
representing the apparent first-order loss rate constant of $NH_3$. An example of the fit is shown
by the black trace in Fig. 3b.
The average $NH_3$ loss rate constant on dew nights was $1.33 \pm 0.5 \times 10^{-4}$ $s^{-1}$ compared to $1.35 \pm$
$0.3 \times 10^{-4}$ $s^{-1}$ on dry nights. In other words, there is no significant difference in the rate of $NH_3$
loss on dew versus non-dew nights. This implies that dew does not actually enhance $NH_3$
deposition under these conditions, suggesting that the aerodynamic and quasi-laminar
resistances dominate over surface resistances. Since $NH_3$ deposition is independent of dew
amount, there could be a large discrepancy between $[NH_4^+]$ for dew on the dewmeter versus
the dew collector if $V_{dew}$ is significantly different on the two surfaces. However, the campaign
averages of $V_{dew}$ on the dewmeter (Table 1) and are within 10% of dew volume obtained off
the collector (data not shown) so $[NH_4^+]$ is likely similar for dew on both platforms.
Since most of the $NH_4^+$ in dew volatilizes and the presence of dew does not affect $NH_3$
deposition overnight, the net impact is a reduction in the overall removal of $NH_3$. As a result,
the atmospheric lifetime and range of $NH_3$ transport will be extended.
**3.4  Potential Influence from Rain Evaporation**
Numerous studies have reported rapid increases of near-surface $NH_3$ within 1-2 h after some
rain events (e.g. Cooter et al., 2010; Walker et al., 2013; Wentworth et al., 2014). Given the
findings discussed in the previous section, one possible explanation is the emission of $NH_3$ from
drying rain droplets. However, unlike dew, some difficult-to-predict fraction of rain will
permeate through the soil thus preventing or delaying the release of $NH_3$. Nonetheless, we
attempt to qualitatively explore this hypothesis by examining the Frac($NH_3$) of four rain
samples collected at RMNP as well as the behaviour of $NH_3$ during rainfall evaporation. Rain
samples were collected with the same procedure used to collect dew, which differs from the
usual method of capturing precipitation via an automated precipitation bucket (e.g. Benedict et
al., 2013a). The precipitation bucket is normally equipped with an O-ring and lid to prevent dry





deposition and dissolution of water-soluble gases when it is not precipitating. On the other hand,
precipitation on the dew collector surface was left exposed and its composition is influenced by
dry deposition and gas-phase dissolution until it was collected at the onset of evaporation.
Supplementary Table S2 gives the concentration of ions measured in rain samples. In general,
concentrations of ions are comparable between dew and rain samples, with the exception of
$NH_4^+$, $SO_4^{2-}$ and $NO_3^-$, which are a factor of 2-4 times more concentrated in rain samples. The
enhancement of these species in rain may reflect additional in-cloud and below-cloud
scavenging of gases ($NH_3$, $HNO_3$ and $SO_2$) and $PM_{2.5}$ aloft. Another possibility is that rain
generally forms during upslope conditions which coincide with more polluted air masses from
east of RMNP, whereas dew typically forms during downslope (cleaner) conditions. Numerous
studies have compared dew composition to rain composition and, in general, have found that
concentrations are enhanced in dew relative to rain (e.g. Polkowska et al., 2008; Wagner et al.,
1992). However, Pierson et al. (1986) reported dew composition to be similar to, but more
dilute than rain at a rural site in Pennsylvania.
Table S3 shows the TOC, IC, TN, pH and calculated Frac($NH_3$) for the four rain samples. Rain
samples were more acidic (average pH = 4.54) than dew samples (average pH = 5.19). The
average Frac($NH_3$) for rain samples was 0.66 suggesting that, on average, roughly two-thirds
of $NH_4^+$ contained in precipitation on surfaces should be liberated as $NH_3$ upon evaporation.
This could pose a significant flux of $NH_3$ to the boundary layer; however, since the fraction of
rain that remains on surfaces after rainfall where it can readily evaporate is not constrained,
only an upper estimate on $NH_3$ fluxes from drying rain can be calculated ($21.2 \pm 13$ ng m$^{-2}$ s$^{-1}$).
This value was calculated in same manner as the dew samples and assumes all rainfall
evaporates.
Figure 6 shows time series of rain accumulation (g m$^{-2}$), air temperature (°C) and $NH_3$ mixing
ratio (ppbv) on four separate days with observed rainfall. The rain accumulation was measured
with the dewfall meter; 1000 g m$^{-2}$ of accumulation is equivalent to 1 mm of rainfall. Rainfall
in excess of 2000 g m$^{-2}$ flooded the collection tray and could not be reliably recorded by the
dewmeter. On 24 June (Fig. 6a) there were three light rainfalls at 15:00, 16:00 and 19:00. The
first event at 15:00 was accompanied by a rapid decrease in $NH_3$ likely due to scavenging by
rain droplets; however, this was not observed for the other two rainfalls that day. For the second
rain event in Fig. 6a (at 16:00) a substantial increase in $NH_3$ (from 0.5 to 1.5 ppbv) was observed
during evaporation and is consistent with $NH_3$ liberation from evaporating rain. However,





evaporation of the other rain events on 24 June (Fig. 6a) as well as those on 27 June (Fig. 6b)
and 11 July (Fig. 6c) are not associated with concomitant increases in $NH_3$, implying that these
rain evaporation events did not release $NH_3$. The evaporation of a more substantial rainfall on
13 August (Fig. 6d) is associated with a temporary rise in $NH_3$ until evaporation ceases at
sundown. The instances of rain evaporation not associated with $NH_3$ increases could be due to
rain with a low Frac($NH_3$), an insignificant amount of $NH_4^+$ in the rain, more atmospheric
dilution than dew mornings due to higher turbulence, and/or significant rain penetration into
the soil.
The results from Fig. 6 are consistent with previous literature showing $NH_3$ increase
immediately following only some rainfall events (Cooter et al., 2010; Walker et al., 2013;
Wentworth et al., 2014). The timing of some rain evaporation events with $NH_3$ increases, as
well as the high Frac($NH_3$) (average = 0.66) of the four measured rain samples suggests it is
possible for rain evaporation from surfaces to be a substantial source of $NH_3$. Neirynck and
Ceulemans (2008) reported $NH_3$ increases concomitant with a drying forest canopy (after
rainfall) as measured by a leaf wetness sensor.
Currently, all $NH_4^+$ collected in precipitation samples is counted towards N deposition.
However, if a fraction of $NH_4^+$ in rainfall is emitted as $NH_3$ during evaporation then N-
deposition could be overestimated. At RMNP, wet deposition of $NH_x$ and dry deposition of
$NH_3$ account for 35% and 18%, respectively, of total reactive nitrogen deposition to the site
(Benedict et al., 2013a). This budget does not take into account any re-emission of $NH_3$ from
drying rain. This budget also does not explicitly account for ammonia uptake or emission during
dew formation and evaporation. A more extensive suite of dew and rainfall measurements is
necessary to quantify the impact of evaporation on annual N-deposition budgets at RMNP.
## 3.5 Implications for other Gases
Other water-soluble gases with similar or lager effective Henry's law constants ($K_H^{eff}$) to $NH_3$
are likely influenced by dew and rain evaporation as well, provided that the relative abundance
of counter-ions allows for volatilization during evaporation. $K_H^{eff}$ is the equilibrium constant
for describing gas-aqueous partitioning and accounts for chemical equilibria in solution. Since
acid-base equilibria are pH dependent, then the $K_H^{eff}$ for acidic and basic species is also pH
dependent (Sander, 2015). $K_H^{eff}$ of $NH_3$ was calculated for the twelve dew samples using data
from Sander (2015) to determine the temperature-dependent Henry's law constant ($K_H$) and





from Bates and Pinching (1950) for the temperature-dependent acid dissociation constant ($K_a$)
of $NH_4^+$ required for the calculation of $K_H^{eff}$. During the study, dew $K_H^{eff}$ spanned two orders
of magnitude and ranged from $4.5 \times 10^5$ to $2.7 \times 10^7$ M atm$^{-1}$. These high values are indicative of
the high water solubility of $NH_3$ at the observed pHs and temperatures. Chameides (1987) used
a simple resistance model to show that deposition of gas-phase species with $K_H^{eff} > 10^5$ M atm$^-$
$^1$ to wetted surfaces (i.e. dew) will be limited by the aerodynamic resistance since the surface
resistance is negligible for such highly water-soluble species. In other words, it is likely that
dew will be a significant night-time sink for other trace gas species with $K_H^{eff} > 10^5$ M atm$^{-1}$
since the dissolution into dew is controlled by aerodynamic processes independent of the
identity of the gas.
Table 1 shows the ratio of [$NH_4^+$] measured in dew to the concentration predicted from
equilibrium calculations using $K_H^{eff}$ and measured $NH_3$ mixing ratio at the onset of evaporation.
The average ratio is low (0.04), consistent with a significant aerodynamic resistance that
prevents $NH_4^+$ saturation in dew droplets overnight.
It has been suggested that dew can act as a reservoir for phenol, nitrophenols, formaldehyde
and HONO based on observations of these species in dew in Santiago, Chile (Rubio et al., 2009,
2012). Zhou et al. (2002) found a correlation between high night-time RH (a surrogate for dew
formation) and HONO increases the following morning coincident with a decrease in RH. A
follow-up study (He et al., 2006) confirmed aqueous solutions mimicking dew can release
>90% of $NO_2^-$ as HONO upon evaporation and observed similar HONO pulses during canopy
drying at a rural forest site in Michigan. Indeed, there is some evidence in the literature that
water-soluble gases (primarily HONO) exhibit a similar behaviour to $NH_3$ during dew
formation and evaporation observed in this study.
Table 2 shows the calculated $K_H^{eff}$ (at 10 °C) for common water-soluble gases that could be
influenced by dew formation/evaporation. This table is by no means exhaustive, but highlights
the important role dew may have as a night-time reservoir and morning source for gases other
than $NH_3$. Formic acid (HCOOH), acetic acid ($CH_3COOH$), nitrous acid (HONO) and nitric
acid ($HNO_3$) all have increasing $K_H^{eff}$ with increasing pH since a more basic solution will
promote dissociation of the acid into its conjugate base. The average pH of dew at RMNP (~5.2)
is likely sufficiently acidic for HONO to experience a surface resistance ($K_H^{eff} \ll 10^5$ M atm$^-$
$^1$) which would limit its transport across the dew-air interface. This is consistent with the low





average $[NO_2^-]$ (0.2 μM) in dew at RMNP, although this might simply reflect low HONO
mixing ratios at the remote RMNP site.
Future field studies on these species should include simultaneous measurements of dew
composition, dew amount, and gas phase mixing ratios to determine whether dew is an
important night-time reservoir and morning source. The latter will be dependent on the fraction
of gas released upon dew evaporation, which requires further investigation specific to each gas.
Based on the findings in this work and Takenaka et al. (2009) it is likely that acidic semi-
volatiles (e.g. acetic acid, formic acid, HONO) will be retained as salts during dew evaporation
at RMNP due to the excess of cations.
**4   Conclusions**
Laboratory experiments involving synthetic dew were performed to determine the factor(s)
controlling the fraction of $NH_4^+$ released as $NH_3$ upon dew evaporation. Results were mostly
consistent with Takenaka et al. (2009) who found that the amount of $NH_3$ that volatilized from
drying aqueous solutions is governed by the relative abundances of $NH_4^+$ and excess "non-
volatile" anions ($\sum$anions - $\sum$cations). However, our findings suggest that acetate, formate and
$HCO_3^-$ should also be counted towards the anion budget. Hence, the Frac($NH_3$) released from
a drying dew sample can be predicted given the ionic composition and pH.
A dewmeter (for dew amount, deployed continuously from 22 June to 31 August) and dew
collector (for dew composition, deployed successfully on 12 occasions) were set up at a remote
field site in Colorado. Dew was relatively dilute compared to previous studies and had an
average $[NH_4^+]$ of 26 μM and pH of 5.2 at sunrise. Simple calculations revealed that dew can
act as a significant night-time reservoir of $NH_3$. At the onset of dew evaporation there was, on
average, roughly twice as much $NH_4^+$ in dew as $NH_3$ in the boundary layer. Furthermore, the
observed $NH_3$ loss overnight was roughly equivalent to amount of $NH_4^+$ that accumulated in
dew by sunrise. Dew composition was used to calculate an average Frac($NH_3$) of 0.94
suggesting that the vast majority of $NH_3$ sequestered in dew overnight is emitted during
evaporation shortly after sunrise. Mornings with dew experience a large increase in $NH_3$
coincident with dew evaporation. Once the dew has completely evaporated, $NH_3$ mixing ratios
either plateau or decrease. Fluxes of $NH_3$ from dew averaged 6.2 ± 5 ng m$^{-2}$ s$^{-1}$ during
evaporation and were calculated using measured $[NH_4^+]$, $V_{dew}$, $t_{evap}$ and Frac($NH_3$). These
fluxes are substantial compared to previously reported fluxes in non-fertilized grasslands
(Wentworth et al., 2014; Wichink Kruit et al., 2007). Mornings without any surface wetness




(neither dew nor rain) never experienced a sharp increase in $NH_3$. Dew-related $NH_3$ fluxes are
likely much more substantial in urban and agricultural areas where $NH_3$ and $[NH_4^+]$ in dew are
significantly higher than at RMNP.
Morning increases of $NH_3$ frequently observed at RMNP (and other sites) are very likely the
result of $NH_3$ emissions during dew evaporation. This hypothesis is supported by: 1) coincident
timing of morning $NH_3$ increases/decreases at the start/completion of dew evaporation, 2) lack
of $NH_3$ morning increase on every non-dew morning, 3) significant $NH_3$ fluxes calculated from
dew, 4) relative abundances of $NH_4^+$ in dew and $NH_3$ in the boundary layer, and 5) approximate
mass balance closure between $NH_3$ lost overnight and $NH_4^+$ accumulated in dew. The
phenomenon of dew "recycling" atmospheric $NH_3$ could lead to an overestimation of $NH_3$ dry
deposition in some ecosystems since dew formed overnight can take up much of the near-
surface ammonia and then release most of it again in the morning upon evaporation.  Such
phenomena are generally not considered in current models of $NH_3$ dry deposition. In addition,
nocturnal loss rates of $NH_3$ were unaffected by the presence of dew. Our results suggest the net
effect of dew is to reduce the overall removal of $NH_3$ and prolong its atmospheric lifetime as
long as the dew composition yields a high Frac($NH_3$).
Similar behaviour (coincident timing of $NH_3$ increases and evaporation) was occasionally
observed for rain. Analysis of four rain samples yielded an average Frac($NH_3$) of 0.66
suggesting $NH_3$ can be released from evaporation of rain in RMNP as well. However, due to
the limited number of samples and lack of constraint for amount of rain sequestered below
ground it is currently impossible to be even semi-quantitative about potential $NH_3$ fluxes from
rain evaporation. This uncertainty merits further research since $NH_x$ wet deposition does not
account for re-release of $NH_3$ from evaporation. Subsequent studies should also examine: 1)
the role of biological processes on surface water composition (e.g. stomatal exchange,
modification via microbes) and 2) influence of guttation (leaf exudate) on surface-air $NH_3$
exchange.
Additional field measurements quantifying $NH_3$ release from dew and rain evaporation are
needed to determine how relevant these phenomena are for modulating $NH_3$ mixing ratios and
N-deposition in different environments (e.g. urban, rural, agricultural). Although the majority
of $NH_4^+$ in dew was released back to the atmosphere at RMNP, this is not necessarily the case
at other locations. For instance, environments with $HNO_3$ deposition exceeding $NH_3$ deposition
to dew would cause a low (or zero) Frac($NH_3$). In addition, a tall canopy can recapture near-





surface $NH_3$ emissions and might modulate emissions from dew drying in the lower canopy
(Walker et al., 2013). Regardless, the ability for dew to act as a morning source of $NH_3$ is
currently absent from atmospheric models, with the exception of a few field-scale models based
on the work of Flechard et al. (1999). The observations from this study suggest dew imparts a
large influence on boundary layer $NH_3$; hence, future work should also focus on developing
model parameterizations for $NH_3$ uptake during dew formation and release from evaporating
dew.
To our knowledge, this is the first study to quantitatively examine the influence of dew on any
water-soluble gas by simultaneously measuring dew amount, dew composition and atmospheric
composition. Although $NH_3$ is the focus of this work, gases with similar $K_H^{eff}$ ($>10^5$ M atm$^{-1}$)
might be influenced by dew formation and evaporation in a comparable manner. Such species
include, but are not limited to, acetic acid, formic acid, HONO and $HNO_3$. Methodology similar
to this study should be used to conduct quantitative field studies for the aforementioned species
to better understand the dynamic influence of dew on boundary layer composition.

## Acknowledgements

The National Park Service (NPS) maintained the field site, provided meteorological data, and
supported the costs of field and laboratory measurements. G. R. W. acknowledges funding from
NSERC and the Integrating Atmospheric Chemistry and Physics from Earth to Space (IACPES)
program for travel funding. The authors wish to thank C. Wallesen, G. Perry and the staff at the
Saddle & Surrey Motel in Estes Park, CO for providing generously discounted rates during
peak tourist season. Lastly, R. Clark and J. Price, and H. Guan provided valuable insight on
construction of the dewmeter and dew collector, respectively.



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





1    Table 1. Total Organic Carbon (TOC), Total Nitrogen (TN), Inorganic Carbon (IC), pH, the

2    ratio of measured to predicted $[NH_4^+]$ in dew, and parameters pertinent to $NH_3$ flux calculations

3    in the field dew samples

| Date | TOC (mg C $L^{-1}$) | IC (mg C $L^{-1}$) | TN (mg N $L^{-1}$) | pH | Frac(NH$_3$) | $V_{dew}$ (mL $m^{-2}$) | $t_{evap}$ (s) | Flux (ng $m^{-2}$ $s^{-1}$) | $[NH_4^+]_{meas:}$ $[NH_4^+]_{eqm}$ |
|------|------|------|------|------|------|------|------|------|------|
| 05/28 | 0.65 | 0.52 | 0.05 | 5.46 | 1.0 | 79.8 | 6000 | 2.4 | 0.02 |
| 06/01 | 2.05 | 1.21 | 0.32 | 5.65 | 0.68 | 97.0 | 6600 | 4.9 | 0.08 |
| 06/23 | 6.10 | 0.58 | 0.61 | 5.35 | 1.0 | 167.2 | 10800 | 7.3 | 0.02 |
| 06/27 | 6.13 | 0.59 | 0.62 | 5.70 | 0.85 | 195.6 | 9000 | 11.0 | 0.05 |
| 06/28 | 9.69 | 0.56 | 0.95 | 5.16 | 1.0 | 161.6 | 8400 | 17.9 | 0.04 |
| 06/29 | 5.27 | 0.19 | 0.46 | 4.83 | 1.0 | 60.9 | 3000 | 7.3 | 0.01 |
| 06/30 | 6.71 | 0.22 | 0.32 | 4.99 | 1.0 | 163.4 | 7800 | 3.3 | 0.01 |
| 07/04 | 6.78 | 0.23 | 1.40 | 5.32 | 1.0 | 206.8 | 16800 | 2.5 | 0.02 |
| 07/19 | 6.53 | 0.11 | 1.47 | 5.85 | 1.0 | 188.2 | 24600 | 1.0 | 0.08 |
| 07/29 | 10.04 | 0.31 | 2.59 | 5.80 | 1.0 | 92.2 | 8400 | 5.4 | 0.09 |
| 08/10 | 7.54 | 0.38 | 0.80 | 5.34 | 1.0 | 96.9 | 7200 | 6.9 | 0.07 |
| 08/11 | 7.28 | 0.17 | 0.85 | 4.67 | 0.74 | 108.4 | 14400 | 4.2 | 0.02 |
| Avg | 6.23 | 0.42 | 0.85 | 5.19 | 0.94 | 134.8 | 10250 | 6.2 | 0.04 |



Table 2. $K_H^{eff}$ of $NH_3$ and other water-soluble gases at 10°C and various pHs

| Gas | pH | $K_H^{eff}$ (M atm$^{-1}$) |
|---|---|---|
| $NH_3$ (ammonia) | 4.5 | $2.1 \times 10^7$ |
| | 6 | $6.7 \times 10^5$ |
| | 7.5 | $2.1 \times 10^4$ |
| HCOOH (acetic acid) | 4.5 | $1.1 \times 10^5$ |
| | 6 | $2.8 \times 10^6$ |
| | 7.5 | $8.9 \times 10^7$ |
| $CH_3COOH$ (formic acid) | 4.5 | $1.9 \times 10^4$ |
| | 6 | $2.3 \times 10^5$ |
| | 7.5 | $7.0 \times 10^6$ |
| HONO (nitrous acid) | 4.5 | $1.3 \times 10^3$ |
| | 6 | $3.9 \times 10^4$ |
| | 7.5 | $1.2 \times 10^6$ |
| $HNO_3$ (nitric acid) | 4.5 | $5.3 \times 10^{12}$ |
| | 6 | $1.7 \times 10^{14}$ |
| | 7.5 | $5.3 \times 10^{15}$ |





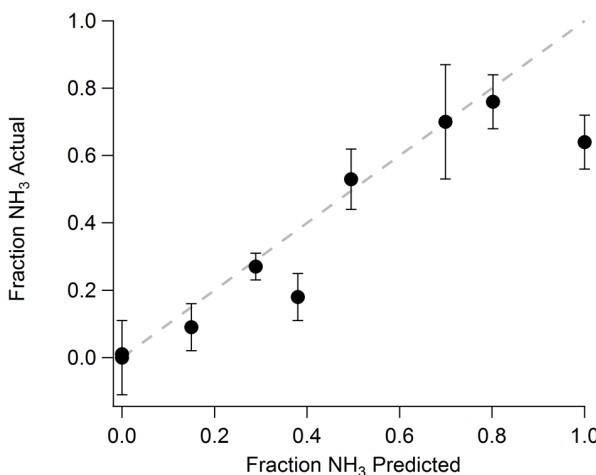

Figure 1. Fraction of NH$_3$ liberated during drying experiments versus the fraction predicted
according to an updated Eq. (2) to include acetate, formate, CO$_3^{2-}$ and HCO$_3^-$ in the anion
balance. Excluding these anions significantly reduces the correlation. Error bars represent $\pm\sigma$
from three experiments per synthetic dew. The dashed line is the 1:1 line.



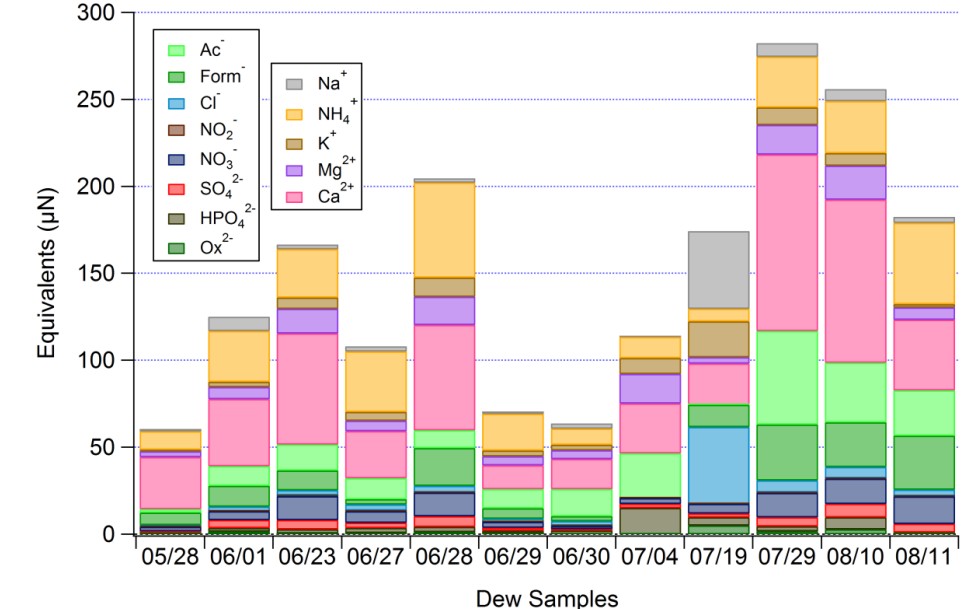

2      Figure 2. Ionic composition (in µN) of ambient dew collected at RMNP.



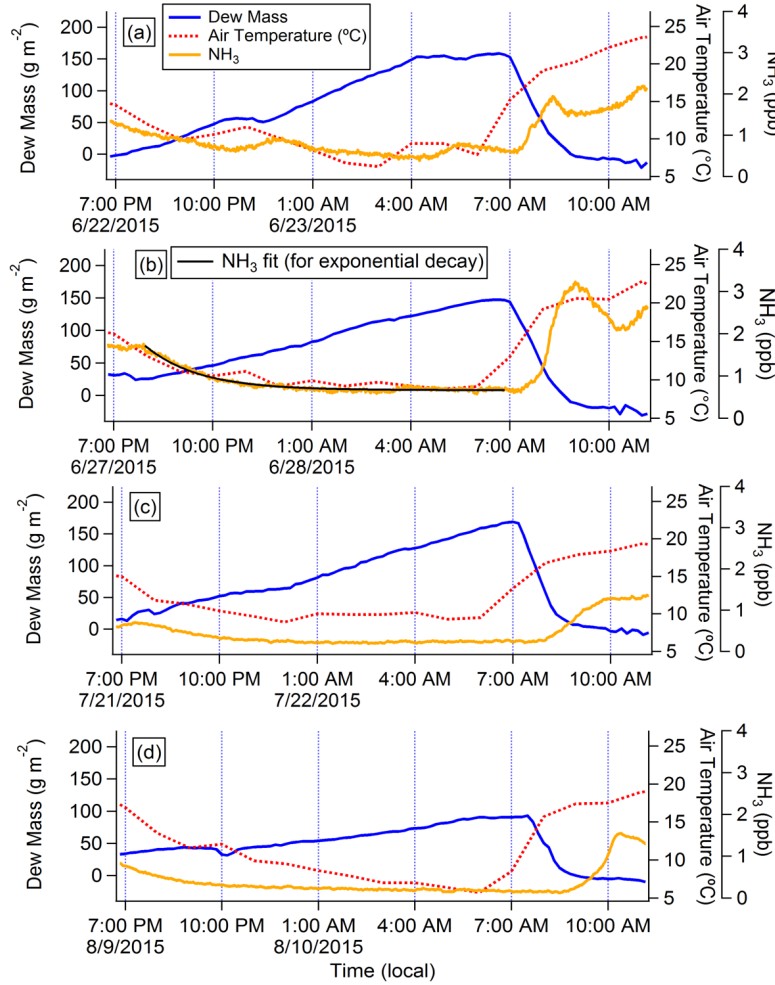

Figure 3. Dew accumulation (blue, g m$^{-2}$), NH$_3$ mixing ratio (orange, ppbv) and air

temperature (red, °C) overnight on a) 22 June, b) 27 June, c) 21 July and d) 9 August 2015.

The black line in (b) is the best fit for the NH$_3$ mixing ratio to an exponential decay function

(see Eq. 4) between 20:00 and the onset of dew evaporation.





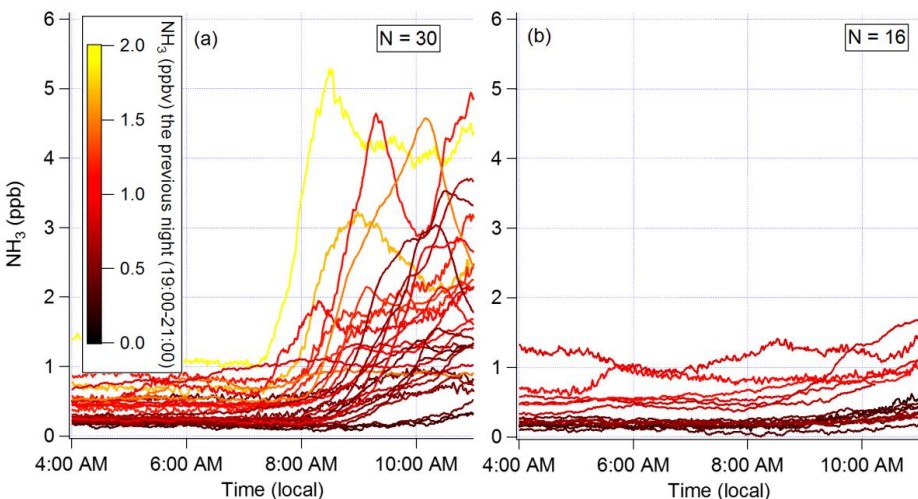

Figure 4. Time series of NH$_3$ mixing ratio (in ppb) from 4:00 to 11:00 on (a) mornings with dew and (b) mornings with no surface wetness. Traces are coloured according to the average NH$_3$ mixing ratio measured the previous night between 19:00 to 21:00.





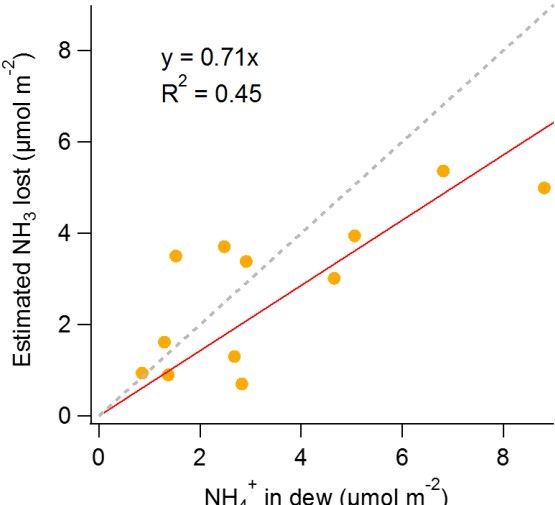

2    Figure 5. Estimated $NH_3$ lost overnight assuming a 150 m boundary layer versus measured

3    $NH_4^+$ accumulated in dew by the onset of evaporation. The red line is the best fit line (forced

4    through the origin) and the dashed grey line is the 1:1 line.





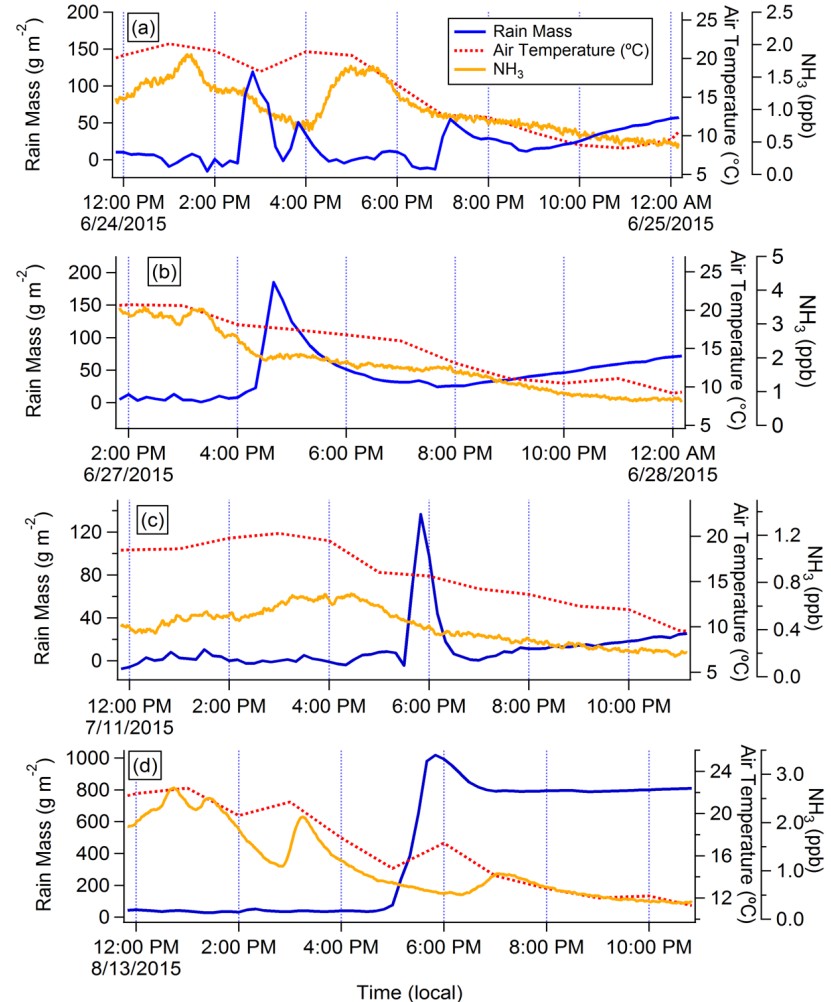

2 Figure 6. Rain accumulation (blue, g m$^{-2}$), NH$_3$ mixing ratio (orange, ppbv) and air

3 temperature (red, °C) during the afternoon and evening on a) 24 June, b) 27 June, c) 11 July

4 and d) 13 August 2015. 100 g m$^{-2}$ is equivalent to 0.1 mm of rain.