# Peer review of "The role of dew as a nighttime reservoir and morning source for atmospheric ammonia"

_Atmospheric Chemistry and Physics, 2016_

## Referee Comment (RC1) · Anonymous Referee #1 · 6 Apr 2016

In this manuscript, the authors presented several lines of evidence in supporting their argument that dew is a nighttime reservoir and a morning source for atmospheric ammonia ($NH_3$). They have demonstrated by the laboratory experiments using artificial dew solutions that the release of ammonia from drying dew can be predicted from dew ion composition. Their field measurement results have shown that ambient $NH_3$ levels decreased in both dry and dew event nights, and then increased in only the mornings following the dew event nights, and the time of release coincided with dew evaporation. Furthermore, the morning increases in $NH_3$ levels can be quantitatively explained by the releases of ammonia in the dews. The laboratory experiments and field measurements were well designed, the results and data were of high quality, and the manuscript is well prepared. I would recommend the publication of this manuscript in Atmospheric Chemistry and Physics. I have several comments need to

be addressed, although they are relatively minor and would not change the general conclusions of this study. Specific comments: 1. NH3 release from drying dew: dry zero air used in the experiment was likely to lower the relative humidity (RH) in the artificial turf to an unreasonable level, and might result in an unrealistically high release fraction. In the real world, the grass canopy surface should be relatively moist because of plant transpiration. One monolayer or more of water could stay on the leaf surface at RH $\geq$40%. The existence of water layers may affect the release of NH3. If dew drying was conducted using zero air at 50% RH, the resulting release fraction might be more realistic. The authors should examine and discuss the potential effect. 2. HCO3- contribution in equation 2: When in equilibrium with atmospheric CO2, the HCO3- concentration is an exponential function of pH. In artificial dew solutions (Table S1), the pH values are mostly higher than 6.35, the pKa1 of H2CO3, and thus HCO3- could be important. However, the average pH of the collected dew samples is 5.19 (page 12, line 24), well below the pKa1 of H2CO3, and thus the contribution from HCO3- to the anion concentration should be negligible. 3. NH3 deposition: It is interesting to note that there was no difference in the average nighttime NH3 loss rate between dew event nights and dry nights. Then the questions become: Was the loss of NH3 due to its deposition to the grass canopy or due to the movement air masses (nighttime down slope flow)? If it was due to its depositional loss, similar amount was released back to the atmosphere in the morning following a dew event night, but not following a dry night; then where was the lost NH3 during a dry night? 4. Ion balance in dew samples: It seems that cations and anions are not in balance in many of the collected dew samples (Figure 2). HCO3- is only $\sim$2 $\mu$M at pH $\sim$5.2, and thus it is unlikely to make up the difference. What could be the missing ions? 5. pH values of the dew and rain samples: It is expected that rainwater to be highly acidic (mostly below pH 5, Table S3), due to high concentrations of NO3- and SO42-, the anions of strong acids, balanced by high concentrations of NH4+, the cation from a weak base (Table S2). It is surprising to see the low pH in the dew samples (Table 1), as the combined equivalents of NO3- and SO42- are lower than those of Ca++, Mg++, K+ and Na+

Please also note the supplement to this comment:
http://www.atmos-chem-phys-discuss.net/acp-2016-169/acp-2016-169-RC1-supplement.pdf
* * *
[Figure]

**Supplement:**

Comments on "The role of dew as a nighttime reservoir and morning source for atmospheric ammonia" by Wentworth et al.

In this manuscript, the authors presented several lines of evidence in supporting their argument that dew is a nighttime reservoir and a morning source for atmospheric ammonia ($NH_3$). They have demonstrated by the laboratory experiments using artificial dew solutions that the release of ammonia from drying dew can be predicted from dew ion composition. Their field measurement results have shown that ambient $NH_3$ levels decreased in both dry and dew event nights, and then increased in only the mornings following the dew event nights, and the time of release coincided with dew evaporation. Furthermore, the morning increases in $NH_3$ levels can be quantitatively explained by the releases of ammonia in the dews. The laboratory experiments and field measurements were well designed, the results and data were of high quality, and the manuscript is well prepared. I would recommend the publication of this manuscript in *Atmospheric Chemistry and Physics*. I have several comments need to be addressed, although they are relatively minor and would not change the general conclusions of this study.

Specific comments:

1. $NH_3$ release from drying dew: dry zero air used in the experiment was likely to lower the relative humidity (RH) in the artificial turf to an unreasonable level, and might result in an unrealistically high release fraction. In the real world, the grass canopy surface should be relatively moist because of plant transpiration. One monolayer or more of water could stay on the leaf surface at RH ≥40%. The existence of water layers may affect the release of $NH_3$. If dew drying was conducted using zero air at 50% RH, the resulting release fraction might be more realistic. The authors should examine and discuss the potential effect.

2. $HCO_3^-$ contribution in equation 2: When in equilibrium with atmospheric $CO_2$, the $HCO_3^-$ concentration is an exponential function of pH. In artificial dew solutions (Table S1), the pH values are mostly higher than 6.35, the pKa1 of $H_2CO_3$, and thus $HCO_3^-$ could be important. However, the average pH of the collected dew samples is 5.19 (page 12, line 24), well below the pKa1 of $H_2CO_3$, and thus the contribution from $HCO_3^-$ to the anion concentration should be negligible.

3. $NH_3$ deposition: It is interesting to note that there was no difference in the average nighttime $NH_3$ loss rate between dew event nights and dry nights. Then the questions become: Was the loss of $NH_3$ due to its deposition to the grass canopy or due to the movement air masses (nighttime down slope flow)? If it was due to its depositional loss, similar amount was released back to the atmosphere in the morning following a dew event night, but not following a dry night; then where was the lost $NH_3$ during a dry night?

4. Ion balance in dew samples: It seems that cations and anions are not in balance in many of the collected dew samples (Figure 2). $HCO_3^-$ is only ~2 µM at pH ~5.2, and thus it is unlikely to make up the difference. What could be the missing ions?

5. pH values of the dew and rain samples: It is expected that rainwater to be highly acidic (mostly below pH 5, Table S3), due to high concentrations of $NO_3^-$ and $SO_4^{2-}$, the anions of strong acids, balanced by high concentrations of $NH_4^+$, the cation from a weak base (Table S2). It is surprising to see the low pH in the dew samples (Table 1), as the combined equivalents of $NO_3^-$ and $SO_4^{2-}$ are lower than those of $Ca^{++}$, $Mg^{++}$, $K^+$ and $Na^+$ in many dew samples (Figure 2).

---

## Referee Comment (RC2) · Anonymous Referee #2 · 9 May 2016

This manuscript examines the potential for ammonia accumulation in dew with subsequent release to the atmosphere during surface drying. The authors examine this process using a combination of laboratory and field dew chemistry measurements, measurement of dew amount, and atmospheric measurements of ammonia. The experiment is well designed and the results and conclusions are, for the most part, well supported by the data, which are of high quality. The manuscript is well written and appropriate for Atmospheric Chemistry and Physics. I recommend publication subject to treatment of the following comments.

Page 3, Line 30: "Most larger scale (regional or global) chemical transport models (CTMs) still employ. . ." The authors should acknowledge that the models are evolving. For example, CTMs commonly used for North America contain a bidirectional framework for NH3 fluxes (see Pleim et al, 2013, doi:10.1002/jgrd.50262; Zhang et al., 2010

doi: 10.1029/2009JD013589).

Rain measurements should be briefly described in Materials and Methods.

Page 13, Line 11: "...NH3 deposited into dew overnight should not necessarily be counted towards the total N-deposition budget...". This is an important statement (and there is a similar statement on Page 14, Lines 25-27) that begs the question, how might the dew measurements differ from real processes in both the grass field and surrounding forest? Is it reasonable to expect that some dew is transferred from the canopy to the ground where the NH4+ would be more likely to remain in the ecosystem? How might the amount and timing of dew in the grass field differ from the surrounding forest?

Regarding the interpretation of the atmospheric measurements of NH3, it is likely that during some periods the emission footprint driving the variability in atmospheric NH3 extends well outside of the grass field in which the dew measurements were made. This may be further complicated by topographically induced advection of NH3 from upslope/downslope as well. For these reasons, and because the field is surrounded by forest, some discussion of the representativeness of the measurements relative to the larger surrounding ecosystem is warranted.

Discussion of NH3 loss rates beginning Page 16, Line 9. The finding of similar loss rates on dew and dry nights is interesting and to me a bit surprising. The implication that dew results in a net lower deposition flux to the ecosystem is important from both a budget standpoint and process modeling. The calculated loss rates, assumed to reflect deposition, are based on a mass balance framework that may be considerably more complicated in complex terrain. For example, the rate could be affected by advection of NH3 depleted air from upslope rather than deposition. In my opinion, this aspect of the paper would benefit from further analysis. The authors should consider including some discussion of meteorological conditions associated with dew versus dry nights. Are wind speed and direction similar? Regarding NH3 deposition processes, I agree with the suggestion that Ra and Rb dominate over the surface resistance at

night. This is another instance where an examination of meteorology may be helpful. Comparison of Ra and Rb on dew versus dry nights would provide some insight into potential differences in exchange processes. Were the CASTNET meteorological measurements active during the study period or were there other measurements from which Ra and Rb may be calculated? If not, even a basic analysis of wind speed and direction during dew versus dry nights would be informative. The results suggest that, assuming the atmospheric resistances are similar on dew versus dry nights, similar rates of non-stomatal deposition occur when the surface is wet versus dry. Can the authors speculate regarding the "dry" process? Of the nights with no surface wetness presented in figure 4b, what were typical maximum values of relative humidity?
* * *

---

## Author Comment (AC1) · 18 May 2016

In this manuscript, the authors presented several lines of evidence in supporting their argument that dew is a nighttime reservoir and a morning source for atmospheric ammonia ($NH_3$). They have demonstrated by the laboratory experiments using artificial dew solutions that the release of ammonia from drying dew can be predicted from dew ion composition. Their field measurement results have shown that ambient $NH_3$ levels decreased in both dry and dew event nights, and then increased in only the mornings following the dew event nights, and the time of release coincided with dew evaporation. Furthermore, the morning increases in $NH_3$ levels can be quantitatively explained by the releases of ammonia in the dews. The laboratory experiments and field measurements were well designed, the results and data were of high quality, and the manuscript is well prepared. I would recommend the publication of this manuscript in *Atmospheric Chemistry and Physics*. I have several comments need to be addressed, although they are relatively minor and would not change the general conclusions of this study.

Specific comments:

1. $NH_3$ release from drying dew: dry zero air used in the experiment was likely to lower the relative humidity (RH) in the artificial turf to an unreasonable level, and might result in an unrealistically high release fraction. In the real world, the grass canopy surface should be relatively moist because of plant transpiration. One monolayer or more of water could stay on the leaf surface at RH ≥40%. The existence of water layers may affect the release of $NH_3$. If dew drying was conducted using zero air at 50% RH, the resulting release fraction might be more realistic. The authors should examine and discuss the potential effect.

Monolayer coverage of water, in a canopy with a leaf area index of 1.5 $m^2$ $m^{-2}$, would represent only 7.6x10$^{-4}$ g $m^{-2}$ of water, more than a factor of a million less water than typical dew volumes, so a relatively minimal amount of ammonia would be associated with this based on bulk solubility. Ammonia could remain at the surface through adsorption of a monolayer coverage of water, in which case the fraction that actually volatilized would be lower than that predicted by the lab experiments. At RMNP the daytime RH was quite low (< 40 %), so the importance of adsorption of $NH_3$ to surface water was likely less significant than it could be at other sites. Subsequent laboratory experiments should investigate this effect by carrying out the drying with air at 50 % RH.

2. HCO3- contribution in equation 2: When in equilibrium with atmospheric CO2, the HCO3- concentration is an exponential function of pH. In artificial dew solutions (Table S1), the pH values are mostly higher than 6.35, the pKa1 of H2CO3, and thus HCO3- could be important. However, the average pH of the collected dew samples is 5.19 (page 12, line 24), well below the pKa1 of H2CO3, and thus the contribution from HCO3- to the anion concentration should be negligible.

We agree with the referee. The $HCO_3^-$ equivalent loading was a non-negligible fraction for the lab dew because the pH of the synthetic dews were mostly above 6.4. On the other hand, ambient dew collected at RMNP was sufficiently acidic to mitigate the importance of $HCO_3^-$ in the ion balance. For reference, the median $[HCO_3^-]$ for ambient dew was 5.4 µM compared to 100 µM for the synthetic laboratory dews.

3. NH3 deposition: It is interesting to note that there was no difference in the average nighttime NH3 loss rate between dew event nights and dry nights. Then the questions become: Was the loss of NH3 due to its deposition to the grass canopy or due to the movement air masses (nighttime down slope flow)? If it was due to its depositional loss, similar amount was released back to the atmosphere in the morning following a dew event night, but not following a dry night; then where was the lost NH3 during a dry night?

Given the approximate mass balance closure (on dew nights) between $NH_3$ lost from the atmosphere and $NH_4^+$ gained in dew, it is likely that deposition is a significant contributor to the observed nighttime loss of $NH_3$. The fact the observed loss rate constant (~0.5 h$^{-1}$) is comparable to literature deposition velocities for $NH_3$ (Schrader and Brümmer, 2014) is further evidence that deposition is a significant contributor to the nocturnal loss of $NH_3$. However, as the reviewer points out, it is not possible to unambiguously associate the nocturnal loss of $NH_3$ with deposition.

On dry nights, the $NH_3$ can deposit to leaf cuticles or to the soil. In the absence of dew, the $NH_3$ may remain adsorbed to the cuticle, sorbed to soil constituents, or dissolved in soil pore water. This has been clarified in the text (page 16, line 13):

"Deposition of $NH_3$ on dry nights could be to either leaf cuticles and/or soil pore water. However, it is not possible to unambiguously attribute the nocturnal $NH_3$ loss solely to deposition. Enhanced downslope flow of cleaner air on dry nights cannot be ruled out as a contributor to nocturnal $NH_3$ loss."

It is possible that dew accumulation prevents or at least lessens deposition to cuticles or soil on dew nights. However, addressing this hypothesis requires further investigation.

4. Ion balance in dew samples: It seems that cations and anions are not in balance in many of the collected dew samples (Figure 2). HCO3- is only ~2 µM at pH ~5.2, and thus it is unlikely to make up the difference. What could be the missing ions?

The missing ions could be longer-chain organic acids (other than acetate and formate) such as succinate, maleate, malonate, and pyruvate. To our knowledge, there are no reported literature values for these species in dew. However, a recent study by Boris et al. (2016) measured the chemical composition of fog water near the ocean and reported average fog water TOC of 17.0 mg C $L^{-1}$ with an average total organic acid concentration of 121 μM, excluding acetate and formate. By comparison, dew at RMNP had an average of 6.23 mg C $L^{-1}$. Organic acids at RMNP could be a result of the oxidation of VOC emissions from the forest surrounding the site.

A second possibility are unmeasured anionic species from wind-blown dust, such as silicates, which could be counter ions for soil mineral cations included in the ion balance ($Ca^{2+}$ and $Mg^{2+}$). Unfortunately, to our knowledge, there are also no constraints on these species in dew.

A brief discussion has been included in the text (page 13, line 3):

"Figure 2 reveals a persistent ion imbalance for ambient dew samples. On average, about 25% more anion is needed to achieve ion balance with the measured cations. This implies that some anions are unaccounted for in the system. Possible explanations include: 1) longer chain organic acids (e.g. succinate, maleate, malonate, and pyruvate) and/or 2) silicates from wind-blown dust."

A future study has been planned to perform lab drying experiments on ambient dew. These subsequent measurements will allow us to: 1) perform a more complete chemical analysis (i.e. for organic acids and silicates) and 2) determine whether ions that are unaccounted for will affect $NH_3$ release from dew evaporation (i.e. evaluate Eq. 2 for ambient samples). Drying experiments could not be done for the RMNP dew since $NH_4^+$ was too dilute to detect $NH_3/NH_4^+_{(residue)}$ after drying and extraction.

5. pH values of the dew and rain samples: It is expected that rainwater to be highly acidic (mostly below pH 5, Table S3), due to high concentrations of NO3- and SO42-, the anions of strong acids, balanced by high concentrations of NH4+, the cation from a weak base (Table S2). It is surprising to see the low pH in the dew samples (Table 1), as the combined equivalents of NO3- and SO42- are lower than those of Ca++, Mg++, K+ and Na+ in many dew samples (Figure 2).

A possible explanation for the acidic pH in ambient dew despite ($2*Ca^{2+} + 2*Mg^{2+} + Na^+ + K^+$) > ($2*SO_4^{2-} + NO_3^-$) is the presence of longer-chain organic acids. In addition, undetected silicates could be significant contributors for the anion balance. The sources and impacts of these species have been discussed in response to the previous comment (#4).

References

Boris, A. J., Lee, T., Park, T., Choi, J., Seo, S. J. and Collett, J. L. Jr.: Fog composition at Baengnyeong Island in the eastern Yellow Sea: detecting markers of aqueous atmospheric oxidations, Atmos. Chem. Phys., 16, 437-453, 2016.

Schrader, F. and Brümmer, C.: Land Use Specific Ammonia Deposition Velocities: a Review of Recent Studies (2004-2013), Water Air Soil Pollut., 225(10), 2114, doi:10.1007/s11270-014-2114-7, 2014.

---

## Author Comment (AC2) · 18 May 2016

This manuscript examines the potential for ammonia accumulation in dew with subsequent release to the atmosphere during surface drying. The authors examine this process using a combination of laboratory and field dew chemistry measurements, measurement of dew amount, and atmospheric measurements of ammonia. The experiment is well designed and the results and conclusions are, for the most part, well supported by the data, which are of high quality. The manuscript is well written and appropriate for Atmospheric Chemistry and Physics. I recommend publication subject to treatment of the following comments.

Page 3, Line 30: "Most larger scale (regional or global) chemical transport models (CTMs) still employ. . ." The authors should acknowledge that the models are evolving. For example, CTMs commonly used for North America contain a bidirectional framework for NH3 fluxes (see Pleim et al, 2013, doi:10.1002/jgrd.50262; Zhang et al., 2010, doi: 10.1029/2009JD013589).

We agree with the referee and have added the following (page 4, line 2):

"However, some recent studies have successfully incorporated a bi-directional $NH_3$ exchange framework into regional and global CTMs (Bash et al., 2013; Wichink Kruit et al., 2012; Zhu et al., 2015)."

Rain measurements should be briefly described in Materials and Methods.

We agree with the referee and have added the following (page 9, line 7):

"If rain had occurred during the night, then rain samples were also collected off of the dew collector in a similar fashion the following morning. Rain samples were unambiguously identified using data from the dewmeter described below."

Page 13, Line 11: ". . .NH3 deposited into dew overnight should not necessarily be counted towards the total N-deposition budget. . .". This is an important statement (and there is a similar statement on Page 14, Lines 25-27) that begs the question, how might the dew measurements differ from real processes in both the grass field and surrounding forest? Is it reasonable to expect that some dew is transferred from the canopy to the ground where the NH4+ would be

more likely to remain in the ecosystem? How might the amount and timing of dew in the grass field differ from the surrounding forest?

The referee raises several good questions that we intend to investigate in subsequent studies. There are several scenarios that could cause discrepancies between dew measurements and ambient dew:

1. If real dew dissolves a substantial amount of salts already present on vegetative surfaces (i.e. particulate matter deposited during the daytime). Since the collector is rinsed prior to deployment, the dew collector will not capture this effect. This would likely cause a change in Frac($NH_3$) and an underestimation of $[NH_4^+]_{dew}$. However, the Frac($NH_3$) will not change if the salt components are non-volatile.
2. If dew is transferred to the ground (as the referee points out). This would cause an overestimation of $V_{dew}$. However, even if dew is transported to the soil surface, it will likely remain at or near the surface and could still be subject to evaporation at sunrise.
3. If there is a large difference in the accumulation/evaporation of dew on different vegetative surfaces (i.e. grass and forest, as the referee points out). The amount and timing of dew depends on a variety of meteorological factors (temperature, RH, wind speed, cloud conditions). Although meteorological factors are likely similar between the grass and surrounding forest, a dense forest canopy could hinder dew formation at the surface and on lower branches due to trapping of IR radiation.

More research is needed to quantitatively explore the impact of these scenarios. We have added a sentence to emphasize the need for additional research (page 13, line 15):

"Additional research is needed to examine the effects of: 1) salts already present on vegetative surfaces on dew composition, 2) dew transfer from leaf to soil prior to evaporation, and 3) different canopies (e.g. forest, tall grass) on the amount and timing of dew accumulation and evaporation."

Regarding the interpretation of the atmospheric measurements of NH3, it is likely that during some periods the emission footprint driving the variability in atmospheric NH3 extends well outside of the grass field in which the dew measurements were made. This may be further complicated by topographically induced advection of NH3 from upslope/downslope as well. For these reasons, and because the field is surrounded by forest, some discussion of the representativeness of the measurements relative to the larger surrounding ecosystem is warranted.

We agree with the referee and have added the following to clarify emission footprint and explicitly discuss the influence of upslope/downslope flow (page 15, line 8):

"It is likely that during some periods the emission/deposition footprint of the atmospheric and dew measurements extends beyond the grassland clearing and into the surrounding forest. While we did not find that the overnight loss rate of ammonia depended on dew amount, the deposition rate of ammonia likely depends on surface type, so estimates of moles of $NH_3$ deposited per $m^2$ from the dew collector may not be representative of the surrounding forest. Upslope and downslope flow conditions could also explain some of the variability in nocturnal $NH_3$ since the latter is prevalent during the nighttime and delivers cleaner air from the west of RMNP.

Subsequent work should be performed to examine the representativeness of grassland dew measurements to the larger surrounding ecosystem."

Discussion of NH3 loss rates beginning Page 16, Line 9. The finding of similar loss rates on dew and dry nights is interesting and to me a bit surprising. The implication that dew results in a net lower deposition flux to the ecosystem is important from both a budget standpoint and process modeling. The calculated loss rates, assumed to reflect deposition, are based on a mass balance framework that may be considerably more complicated in complex terrain. For example, the rate could be affected by advection of NH3 depleted air from upslope rather than deposition. In my opinion, this aspect of the paper would benefit from further analysis.

We agree that the nocturnal $NH_3$ loss rate is affected by more than just deposition, and could be affected by upslope/downslope advection as the referee suggests. Therefore, we have added an additional point to our list of assumptions (page 15, line 25):

"…and 4) no influence from horizontal advection (i.e. upslope/downslope flow) on $NH_3$"

The authors should consider including some discussion of meteorological conditions associated with dew versus dry nights. Are wind speed and direction similar?

The average nocturnal wind speed on dew nights was less than dry nights (1.3 m s$^{-1}$ versus 2.2 m s$^{-1}$). On the other hand, average wind direction was from the NW for both dew nights (307°) and dry nights (313°) indicating downslope flow in both instances. The average maximum nocturnal RH on dew nights was 75%, significantly higher than on dry nights (53%).

Regarding NH3 deposition processes, I agree with the suggestion that Ra and Rb dominate over the surface resistance at night. This is another instance where an examination of meteorology may be helpful. Comparison of Ra and Rb on dew versus dry nights would provide some insight into potential differences in exchange processes. Were the CASTNET meteorological measurements active during the study period or were there other measurements from which Ra and Rb may be calculated? If not, even a basic analysis of wind speed and direction during dew versus dry nights would be informative. The results suggest that, assuming the atmospheric resistances are similar on dew versus dry nights, similar rates of non-stomatal deposition occur when the surface is wet versus dry.

Unfortunately $R_a$ and $R_b$ cannot be reliably calculated during the study since there was no instrumentation with which to measure friction velocity. As stated above, winds were calmer on dew nights while the average wind directions were similar. We agree with the referee that this is important to discuss and have added the following to the manuscript (page 16, line 13):

"The average nocturnal wind speed on dew nights was lower than on dry nights (1.3 m s$^{-1}$ versus 2.2 m s$^{-1}$). Lower wind speeds typically result in a higher $R_a$ and $R_b$. It is possible that increased aerodynamic and quasi-laminar resistances on dew nights are partially compensated for by a lower surface resistance due to dew, such that the overall canopy resistance is similar on dew nights and dry nights. Average nocturnal wind direction was from the NW (i.e. downslope flow) on both dew nights (307°) and dry nights (313°). The average nocturnal maximum for RH was

75% on dew nights and only 53% on dry nights. The lower wind speeds and higher RH on dew nights are consistent with the meteorological conditions favourable for dew formation."

Can the authors speculate regarding the "dry" process? Of the nights with no surface wetness presented in figure 4b, what were typical maximum values of relative humidity?

See our above response for the typical maximum RH values. Deposition on dry nights could be either through adsorption to the leaf cuticle or soil constituents or through dissolution into soil pore water. It is also possible that enhanced downslope flow on dry nights is partially responsible for nocturnal $NH_3$ loss at RMNP. The following has been added to the manuscript (page 16, line 13):

"Deposition of $NH_3$ on dry nights could be to either leaf cuticles and/or soil pore water. However, it is not possible to unambiguously attribute the nocturnal $NH_3$ loss solely to deposition. Enhanced downslope flow of cleaner air on dry nights cannot be ruled out as a contributor to nocturnal $NH_3$ loss."

**References**

Bash, J. O., Cooter, E. J., Dennis, R. L., Walker, J. T. and Pleim, J. E.: Evaluation of a regional air-quality model with bidirectional $NH_3$ exchange coupled to an agroecosystem model, Biogeosciences, 10(3), 1635–1645, doi:10.5194/bg-10-1635-2013, 2013.

Wichink Kruit, R. J., Schaap, M., Sauter, F. J., Van Zanten, M. C. and van Pul, W. A. J.: Modeling the distribution of ammonia across Europe including bi-directional surface-atmosphere exchange, Biogeosciences, 9(12), 5261–5277, doi:10.5194/bg-9-5261-2012, 2012.

Zhu, L., Henze, D., Bash, J., Jeong, G.-R., Cady-Pereira, K., Shephard, M., Luo, M., Paulot, F. and Capps, S.: Global evaluation of ammonia bi-directional exchange and livestock diurnal variation schemes, Atmos. Chem. Phys., 15, 12823–12843, doi:10.5194/acp-15-12823-2015, 2015.